# Identifying the Leverage Points in the Household Solid Waste Management System for Harare, Zimbabwe, Using Network Analysis Techniques

Phyllis Rumbidzai Kwenda [1,*], Gareth Lagerwall [1], Sibel Eker [2] and Bas van Ruijven [2]

[1] Department of Agricultural Engineering, School of Engineering, University of KwaZulu-Natal, Carbis Road, Scottsville, Pietermaritzburg 3201, South Africa

[2] International Institute of Applied System Analysis (IIASA), Schlossplatz 1, A-2361 Laxenburg, Austria

* Correspondence: phyllisrumbidzaikwenda@gmail.com

**Abstract:** Managing household solid waste (HSW) has gone beyond what the Harare local government can handle. Inadequate knowledge of the interactions existing between issues that affect the efficient running of waste management systems is one of the major hindrances in waste management planning in developing countries like Zimbabwe. The complexity of the waste management system for a given municipal area needs to be identified and understood to generate appropriate and efficient waste management strategies. Network analysis (NA) is a methodology extensively used in research to help reveal a comprehensive picture of the relationships and factors related to a particular phenomenon. The methodology reduces the intricacy of large systems such as waste management to smaller and more understandable structures. In this study, NA, which was done mainly using the R software environment, showed a result of 1.5% for network density, thus signifying that for Harare, waste management strategies need to be 'seeded' in various parts of the system. The Pareto principle and the 3Rs (Reduce, Reuse, Recycle) concept were applied to suggest the issues to prioritize and generate strategies that could potentially affect significant change to the city's waste management system. The key issues identified, in their order of importance, are an increase in uncollected waste, low waste collection efficiency, increase in illegal waste dumping, the deteriorating country's economy, reduced municipal financial capacity, reduced municipal workforce capacity, inadequate or unreliable waste data, increase in waste volume, increase in the number of street vendors, no waste planning and monitoring unit, no engineered landfills in the city, increase in waste collection pressure, low waste collection frequency, increase in the unemployment rate, reduced municipal technical capacity, few waste collection vehicles, limited vehicles maintenance, distinct socio-economic classes, high vehicles breakdown, and increase in population.

**Keywords:** household solid waste management; network analysis; leverage points; PAJEK; VOSviewer; R; Harare; Zimbabwe

## 1. Introduction

About 33% of the approximately 2.01 billion tonnes of MSW produced per year globally is managed in a way that is not environmentally friendly [1]. With the world urbanizing at an unprecedented rate, municipalities ought to constantly evolve to ensure sustainability [2–4]. Decision making in waste management should account for uncertainty and the complexity of the system [5]. Moreso, intensive considerations should be placed on the waste management system structure choices and their implications on environmental and financial costs [6].

During the development of its national solid waste management plans, Zimbabwe currently uses the SWOT analysis to leverage or assess strengths, weaknesses, opportunities, and threats in waste management. However, due to the complexity of waste management systems, municipalities ought to adopt the national solid waste management plan to make

them more specific and efficient for individual municipalities. The SWOT analysis is similar to Network Analyses (NA) in that it creates a visual representation of factors most likely to influence municipalities and involves different perspectives from different stakeholders, it has some limitations which can be overcome by using NA. Examples of these limitations are that some factors can be overlooked and the input of some factors in the analysis can be biased. Retaining the use of SWOT analysis at the national level and adopting the SWOT analysis at the municipal level can aid in developing tailor-made strategies while maximizing the benefits of both analyses.

The NA methods have been widely used in waste management research [7–11]. They help better understand the role, driving force, and links among various actors, thus identifying bottlenecks that affect efficient strategic planning and implementation [7,12–14]. The NA technique explores the patterns and structure of network relationships over time [9]. Various software can be used to analyze networks such as NetDraw, Pajek, StOCHNET, STRUCTURE, UCINET, VOSviewer, and RStudio [15]. However, this study used the freely available and user-friendly network analysis programs Pajek, RStudio, and VOSviewer.

Generally, waste management strategy development is guided by the 3Rs (Reduce, Reuse and Recycle) policy approach for the sustainable management of waste [16]. As illustrated in Figure 1 below, the 3Rs offer an environmentally friendly option to deal with waste through Reduction, Reuse, and Recycling, in order of priority [17].

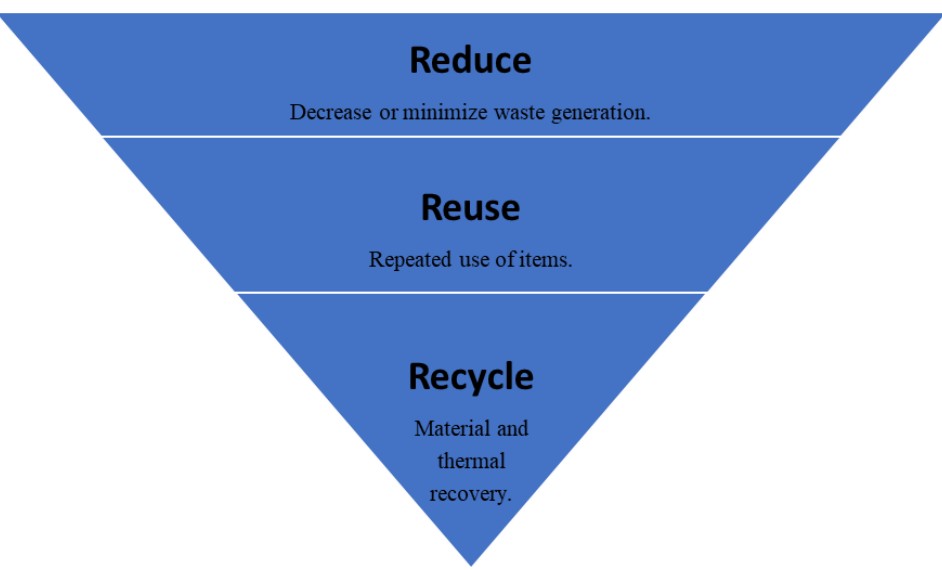

**Figure 1.** The 3Rs policy for sustainable waste management.

Additionally, the 3Rs policy distributes the responsibility of waste management among different stakeholders like the municipality, communities, and private organizations [18]. Incorporating this policy in waste management not only creates jobs but also minimizes illegal waste dumping, saves landfill space, and can generate additional revenue for the municipality or city residents [19]. Although it is widely applied in developed countries, this policy is difficult to use in developing countries due to technical, financial, and expertise constraints. This study attempted to, in conjunction with Network Analysis (NA) techniques, apply the 3Rs policy to Harare's waste management system.

The NA was used to visualize and quantify relationships between HSW issues in Harare. Consequently, the potential leverage points in the city's waste management system were identified. The objectives driving the study were to (a) identify the relationships that exist between HSW issues in Harare, (b) develop a network and calculate network statistics, (c) use the Pareto principle to identify the top 20 most important HSW issues, and (d) apply the 3Rs principle to suggest the potential leverage points tom significantly improve waste management in the city.

## 2. Methodology

The description of the terminology used in this study is summarized in Table A2 in Appendix A. Data used in developing Harare's HSW management issues network were obtained from a literature study conducted by Kwenda, et al. [20]. The HSW management issues (listed in Table A3 in Appendix A) and the relationships between these issues identified by Kwenda, Lagerwall, Eker, and Van Ruijven [20] are the network nodes and edges shown in this study, respectively. To analyze the data using PAJEK [21], RStudio [22], and VOSviewer [23], both the *.xlsx* and *.net* network data files were needed.

A *.txt* file named HHSWM_RData was initially created in Notepad. It summarized all the HSW management issues in pairs showing the 'driver' first separated from the 'outcome' with space (using the Tab key). A 'driver' is a node causing a change in another (the 'outcome'). In this study, the issues considered as 'drivers' are those that have a greater out-degree (i.e., are causing changes to many other nodes) than in-degrees (i.e., than other nodes influencing them). The study hypothesizes that the greater the degree of a given node, the more influential it is in the network and thus should be prioritized when developing waste management strategies, as summarized in Figure 2 below. In these files, the issues were, however, represented as numbers, as shown in Table A3 in Appendix A, to enable better visualization in the networks. The TXT2PAJEK software (Technical University of Munich 2020) was then used to convert the HHSWM_RData from a *.txt* to a *.net* file. Network data in PAJEK and VOSviewer software was imported in the *.net* format, while the *.xlsx* file format developed in Microsoft Excel was used for analysis in R.

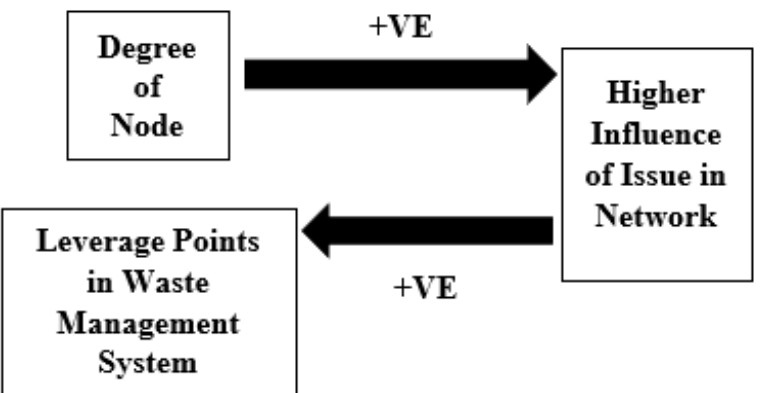

**Figure 2.** Hypothesis of the research study.

A network is defined as a set of interactions or relationships among different entities [24,25]. As a preliminary study and to visualize the complexity, a network was developed in PAJEK. After importing the HHSWM_RData.*net* file in PAJEK, a network in simple circular format was generated. The PAJEK software was developed based on prior experience with "graph data structure and algorithms libraries Graph and X-graph, collection of network analysis and visualization programs STRAN, RelCalc, Draw, Energ, and SGML-based graph description markup language NetML" [21]. To identify the clusters within the network, HHSWM_RData.*net* was imported and run in VOSviewer, which automatically lists the different clusters within the network.

For the main study and to calculate network statistics, RStudio was used. The HHSWM_RData.*xlsx* file was imported into RStudio. The commands in Table A4 in Appendix A were then entered into the RStudio console panel to generate networks and calculate network statistics such as network size, edges, number of clusters, and density. Among other methods that can be used to identify the most influential actors in a network, this study used the most used methods namely degree centrality and betweenness centrality [25–28]. Network statistics are numerical values that are calculated and used to describe essential properties of parts or the entire network [29].

## 3. Results and Discussion

In the preliminary study, Harare's HSW management system network in the Fruchterman–Reingold layout was generated using PAJEK as shown in Figure A1 in Appendix A. The many edges and loops observed suggest that Harare's HSW management phenomenon is rather complex. Complex systems are entities characterized by the interaction of multiple components [30,31]. They are often difficult to understand and forecast as they are usually multifaceted and dynamic [32–34]. Network statistics are, however, needed to estimate the extent of network complexity as was done during the main study using R.

Figure 3a,b below shows networks developed in R based on each node's degree and betweenness centrality values, respectively. The greater the degree of centrality of a node, the greater the number of nodes linked to that given node [24–26]. Degree centrality is an easy method of identifying influential actors in a network [25,27]. On the other hand, nodes with high betweenness centrality have greater power to influence other actors and can connect clusters within a network [26]. These are key in the flow of change or influence within the network. Addressing issues with high degree centrality values will simultaneously address many other issues within a network cluster, while those with higher betweenness centrality values have the power to connect clusters [26]. As shown in Table 1 below, the difference in the order of issues when ranked using degree centrality values as opposed to betweenness centrality values suggests that how connected an issue is to others does not reflect how much influence it has in the network.

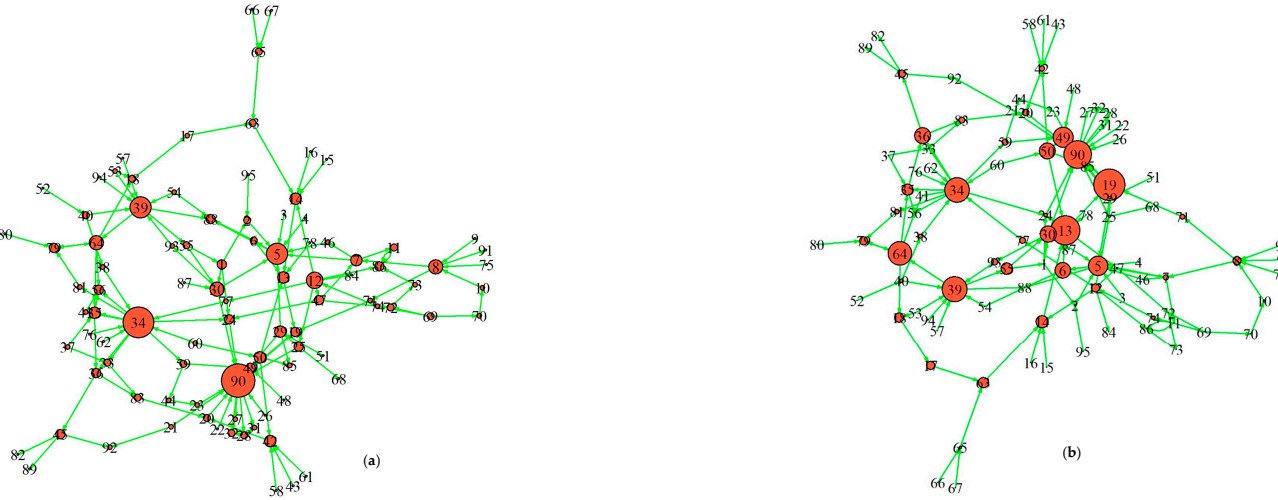

**Figure 3.** Network developed in RStudio with node size calculated from node degree centrality (**a**), and betweenness centrality (**b**).

The size of the network was 95, the number of edges was 138, the network density was 1.5%, and the number of clusters was 15. The low network density of 1.5% suggests that for significant change to be observed in Harare's HSW management system, solutions would need to be 'seeded' in various parts of the network. Moreso, the greater the number of clusters within the network, the greater the need for strategies to be spread across the network to observe significant changes to the HSW management system.

The Pareto principle, also known as the 80:20 rule, states that fewer causes can be responsible for most effects within systems [35–40]. This principle can be key in improving the decision-making process as it helps identify the key problems to prioritize [39,41,42]. The Pareto principle has been applied in waste management decision-making [40,43,44]. The principle suggests that prioritizing solving the top 20% of the key issues in an organization will yield the most significant change within the system. This principle was applied in this study, and the top 20 key issues identified in the city's waste management system are presented in Table 1 above.

**Table 1.** Top 20 HSW management issues in Harare in descending order of influence based on degree and betweenness centrality, respectively.

|  | **Degree Centrality** | **Betweenness Centrality** |
|---|---|---|
| 1 | Illegal waste dumping (14) | Uncollected waste (337.5) |
| 2 | Financial capacity (13) | Waste collection efficiency (316) |
| 3 | Economy (9) | Illegal waste dumping (300) |
| 4 | Waste volume (9) | Economy (269) |
| 5 | Socio-economic class (7) | Financial capacity (268) |
| 6 | Population (6) | Workforce capacity (253) |
| 7 | Street vendors (6) | Waste data (217) |
| 8 | Workforce capacity (6) | Waste volume (210) |
| 9 | Consumption (5) | Street vendors (181) |
| 10 | Waste collection efficiency (5) | Planning and monitoring (176) |
| 11 | Waste collection frequency (5) | No engineered landfills (173.5) |
| 12 | Perceptions (5) | Waste collection pressure (170) |
| 13 | Technical capacity (5) | Waste collection frequency (135.5) |
| 14 | Waste recovery (5) | Unemployment rate (126) |
| 15 | Planning and monitoring (5) | Technical capacity (124) |
| 16 | Uncollected waste (4) | Waste collection vehicles (117) |
| 17 | Waste data (4) | Vehicles maintenance (102) |
| 18 | No engineered landfills (4) | Socio-economic class (91.5) |
| 19 | Worker strikes (4) | Vehicles breakdown (91) |
| 20 | Access control (4) | Population (90) |

To observe how the most influential issues (those ranked according to betweenness centrality values) are placed within the clusters that make up the network, Table 2 below was developed. The table shows the HSW management issues in each of the 15 clusters in the network, while the most influential issues are in red. At least one key issue is in each of the clusters except clusters 11, 13, and 14. This is ideal and hence agrees with the interpretation of the low network density of 1.5%, which suggests that various strategies be 'seeded' within the network for effective change to be observed in the entire system. These results thus suggest that the city of Harare prioritizes developing strategies that address these 20 issues to effect significant change, as these are its key leverage points within the system.

To identify the major drivers and outcomes among Harare's HSW management key issues (those ranked according to betweenness centrality values), their in-degree and out-degree values given in Table A1 in the Appendix A section were compared. The issues with higher in-degree centrality as compared to out-degree centrality were classified as network outcomes, while the vice-versa were classified as network drivers, as shown in Table 3. Drivers are issues that influence other issues more than they were influenced, while outcomes are vice versa. This information is important in developing strategies as in addressing a driver; the focus is placed on the driver e.g., to improve financial capacity, solutions are built around making funds available. In contrast, for an outcome like illegal waste dumping, it would require that the factors contributing to this issue, e.g., uncollected waste or street vendors, among others, be identified and addressed. Issues that qualify as both drivers and outcomes have the same in and out-degree centrality values. Addressing issues like these, for example, uncollected waste, would require that the strategies developed to answer the questions 'Why are we having uncollected waste remaining in our residential areas?' as well as 'What can we do to reduce the amount of uncollected waste?'. As a result, strategy development approaches applicable to both outcomes and drivers need to be applied in these cases.

**Table 2.** Clusters in Network identified using VOSviewer.

| Unemployment | Cluster 2 (9) | Cluster 3 (8) |
|---|---|---|
| 11 Changes in waste consumption patterns | 34 Financial capacity | 1 Poverty |
| 3 Operation Murambatsvina | 41 Municipal expansion | 2 Urban agriculture |
| 4 Festive season | 56 Municipal capacity | 24 Receptacle availability |
| 46 Waste generation rate | 62 Loans | 30 Street vendors |
| 47 Disposable income | 64 Workforce capacity | 55 Unemployment rate |
| 5 Waste volume | 76 Corruption | 87 HIV/AIDS epidermic |
| 7 Consumption | 79 Worker strikes | 93 National informal economy |
| 72 Black access to goods and services | 80 PPE | 95 Weather |
| 73 Processed products | 81 Salaries | |
| 86 Changes in waste composition patterns | | |

| Cluster 4 (8) | Cluster 5 (7) | Cluster 6 (7) |
|---|---|---|
| 13 Waste collection efficiency | 17 Vehicles breakdown | 25 Waste management knowledge |
| 39 Economy | 18 Vehicles maintenance | 29 Perceptions |
| 54 Hyperinflation | 53 Foreign currency | 50 Planning and monitoring |
| 57 Multicurrency | 63 Waste collection vehicles | 51 Gender |
| 6 Waste collection pressure | 65 Fuel | 60 Billed properties |
| 78 Communication | 66 Fuel cost | 68 Community participation |
| 88 Cost of service provision | 67 Distance from the disposal site | 85 Strategy implementation |
| 94 National formal economy | | |

| Cluster 7 (7) | Cluster 8 (7) | Cluster 9 (6) |
|---|---|---|
| 10 Urbanization | 22 Waste burying | 19 Uncollected waste |
| 12 Socio-economic class | 26 Home ownership | 71 Illegal settlements |
| 69 Colonization | 27 Mosquito and rodent manifestation | 75 Capita city |
| 70 Travel regulations | 28 Sewer blockages | 8 Population |
| 74 Racial segregation | 31 Environmental pollution | 9 Natural population increase |
| 77 Waste collection fee | 32 Health risk | 91 Population density |
| 84 Waste transfer to poorer suburbs | 90 Illegal waste dumping | |

| Cluster 10 (6) | Cluster 11 (5) | Cluster 12 (5) |
|---|---|---|
| 20 Crude waste dumping | 21 Open waste burning | 23 Waste law enforcement |
| 33 Sanitary landfilling operations | 45 Access control | 44 Waste policy |
| 35 Technical capacity | 82 Fights | 48 Weighbridge at the dumpsite |
| 36 No engineered landfills | 89 Accidents | 49 Waste data |
| 37 Landfill equipment | 92 Dumpsite fires | 59 Research |
| 83 Sanitary formal solid waste disposal | | |

| Cluster 13 (4) | Cluster 14 (3) | Cluster 15 (3) |
|---|---|---|
| 42 Waste recovery | 38 Donor funding | 14 Waste collection frequency |
| 43 Waste separation at source | 40 Political instability | 15 Harare waste collection policy |
| 58 Private companies | 52 Land reform program | 16 Suspension of waste collection services |
| 61 Recyclers | | |

The potential impact areas of policy change in this study are summarized in Table 4 below. The 3Rs concept was applied to the waste management system in Harare while prioritizing the key issues summarized in Table 3 above.

**Table 3.** Classification of key HSW management issues into either drivers or outcomes.

|  | Key HSW Management Issues | Driver/Outcome |
|---|---|---|
| 1 | Uncollected waste | Driver/outcome |
| 2 | Waste collection efficiency | Outcome |
| 3 | Illegal waste dumping | Outcome |
| 4 | Economy | Outcome |
| 5 | Financial capacity | Driver |
| 6 | Workforce capacity | Driver |
| 7 | Waste data | Outcome |
| 8 | Waste volume | Outcome |
| 9 | Street vendors | Driver/outcome |
| 10 | Planning and monitoring | Driver |
| 11 | No engineered landfills | Driver/outcome |
| 12 | Waste collection pressure | Driver |
| 13 | Waste collection frequency | Outcome |
| 14 | Unemployment rate | Driver |
| 15 | Technical capacity | Outcome |
| 16 | Waste collection vehicles | Outcome |
| 17 | Vehicles maintenance | Outcome |
| 18 | Socio-economic class | Driver |
| 19 | Vehicles breakdown | Driver/outcome |
| 20 | Population | Driver/outcome |

Generally, although most developing countries are making efforts, most of their municipalities are not able to efficiently manage the increasing solid waste volumes [45]. With the increase in population, urbanization, and standard of living, among other things, Zimbabwe is no exception. However, due to municipal incapacitation mostly due to limited funds and expertise, some of the strategies suggested in Table 4 above might be better adopted as long-term plans for the city of Harare [20]. For example, increasing waste collection frequency, increasing waste collection capacity, building sanitary engineered landfills, establishing a planning and monitoring department and increasing technical capacity. Other strategies, such as taxing companies producing waste-generating products, require time to establish and enforce while decreasing the number of street vendors might prove difficult given the economic crisis in the country. However, reduction in waste generation rate, waste recovery, and privatizing of some waste management operations are strategies that future studies can assess for adaptation as a short-term response to the household solid waste management crisis. These can potentially reduce waste volume, uncollected waste, the amount of workforce needed, waste collection trips, waste collection pressure, and illegal waste dumping, thus addressing most of the key leverage points within the waste management system as identified through NA.

**Table 4.** Potential application of the 3Rs policy in Harare's waste management system.

| Waste Generation | Waste Collection and Transport | Waste Disposal |
|---|---|---|
| **Reduce waste volume.**<br><br>- **Avoid unnecessary consumption thus reducing the waste generation rate.**<br>- **Introduce taxes on all production companies equating to the amount of waste generated by consumers using their products. The taxes will be used to fund initiatives to recover this waste.** | ■ Lower the number of waste collection trips.<br>   - Increase waste collection efficiency by (a) improving planning and monitoring e.g on issues like vehicle routing efficiency. However, this would entail that proper waste data collection methods would need to be employed to (b) increase waste collection capacity by improving waste vehicle maintenance and repairs, and (c) increase waste collection frequency by adjusting the city's waste collection policy.<br><br>■ Minimize the amount of uncollected waste.<br>   - Increase waste collection efficiency which can be improved by increasing waste collection capacity.<br>   - Waste collection capacity can be increased by increasing the number of waste collection vehicles available. This can be done by reducing the number of vehicle breakdowns by offering improved vehicle maintenance and improving the financial capacity of the council to procure more vehicles.<br><br>■ Lower the amount of workforce required. (Instead of working on increasing the workforce which it might not afford to, the local authority can work on reducing it by diverting some responsibilities to external companies thus ensuring efficiency in operations)<br>   - This can be done by contracting other private waste collection service providers as well as vehicle maintenance companies to service the waste management fleet.<br><br>■ Lessen waste collection pressure<br>   - This can be done by encouraging waste recovery through **recycling, reuse**, and or composting at the household or community level. This then will reduce the amount of waste the city council needs to collect. | ■ Minimize illegal waste dumping.<br>   - Work on building a sanitary engineered landfill.<br>   - Increase the technical and financial capacity of the local authorities to be able to conduct sanitary waste disposal operations.<br>   - Lessen the number of illegal street vendors which in turn minimizes the disposal of waste in undesignated areas.<br>   - Increase **waste recovery and recycling** at the Pomona dumpsite |

## 4. Conclusions

The small network density value of 1.5% shows that to effect significant change to the HSW management system in Harare. Strategies must be 'seeded' in various parts of the waste management system i.e., waste generation, collection, and disposal. The top 20% key leverage points identified in Harare's waste management system are, in their order of importance, increase in uncollected waste, low waste collection efficiency, increase in illegal waste dumping, the deteriorating country's economy, reduced municipal financial capacity, reduced municipal workforce capacity, inadequate or unreliable waste data, increase in waste volume, increase in the number of street vendors, no waste planning and monitoring unit, no engineered landfills in the city, increase in waste collection pressure, low waste collection frequency, increase in the unemployment rate, reduced municipal technical capacity, few waste collection vehicles, limited vehicles maintenance, distinct socio-economic classes, high vehicles breakdown, and increase in population. Applying the 3Rs concept to address these issues would require the municipality to develop strategies to reduce waste volume during waste generation and encourage waste recovery or reuse during waste collection and disposal. Among other possible strategies suggested in this study for Harare's HSW management system, reducing waste generation rate, waste recovery, and privatizing some waste management operations are potential short-term solutions that future studies can assess for implementation. These are likely to reduce waste volume, uncollected waste, the workforce needed, waste collection trips, waste collection pressure, and illegal waste dumping, thus addressing most of the key leverage points within the city's waste management system. To develop national solid waste management plans, the city of Harare currently uses the SWOT analysis technique. However, despite its advantages which are similar to NA analysis, this method carries limitations, such as the input of some factors in the analysis can be biased. This limitation can be overcome by the concurrent use of NA analysis, as in this study, when developing local municipal solid waste management plans.

## 5. Future Recommendations

Future research can use the network developed in this study to formulate a Causal Loop Diagram and, ultimately, a system dynamics (SD) model that simulates the HSW management system in Harare. The model can then be used for further decision-making and to eliminate the limitation of the NA and SWOT analysis methods so that their findings can be projected into the future.

**Author Contributions:** Conceptualization, P.R.K., G.L., S.E. and B.v.R.; methodology, P.R.K.; software, P.R.K.; validation, P.R.K., G.L., S.E. and B.v.R.; formal analysis, P.R.K.; investigation, P.R.K.; resources, G.L.; data curation, P.R.K.; writing—original draft preparation, P.R.K.; writing—review and editing, P.R.K., G.L., S.E. and B.v.R.; visualization, P.R.K.; supervision, G.L., S.E. and B.v.R.; project administration, P.R.K. and G.L.; funding acquisition, G.L. All authors have read and agreed to the published version of the manuscript.

**Funding:** This work was funded by the National Research Foundation/South African Systems Analysis Centre (NRF/SASAC) [grant numbers 132921].

**Institutional Review Board Statement:** Not applicable.

**Informed Consent Statement:** Not applicable.

**Data Availability Statement:** Not applicable.

**Conflicts of Interest:** The authors declare no conflict of interest.

## Appendix A

**Table A1.** Degree and betweenness centrality of nodes in the network.

| Node | Degree Centrality | | | Betweenness Centrality | Node | Degree Centrality | | | Betweenness Centrality |
|---|---|---|---|---|---|---|---|---|---|
| | Degree | In-Degree | Out-Degree | | | Degree | In-Degree | Out-Degree | |
| 1 "Poverty" | 4 | 1 | 3 | 20 | 49 "Waste data" | 4 | 3 | 1 | 217 |
| 2 "Urban agriculture" | 3 | 2 | 1 | 19.5 | 50 "Planning and monitoring" | 5 | 2 | 3 | 176 |
| 3 "Operation Murambatsvina" | 1 | 0 | 1 | 0 | 51 "Gender" | 1 | 0 | 1 | 0 |
| 4 "Festive season" | 1 | 0 | 1 | 0 | 52 "Land reform program" | 1 | 0 | 1 | 0 |
| 5 "Waste volume" | 9 | 7 | 2 | 210 | 53 "Foreign currency" | 2 | 0 | 2 | 0 |
| 6 "Waste collection pressure" | 3 | 1 | 2 | 170 | 54 "Hyperinflation" | 2 | 0 | 2 | 0 |
| 7 "Consumption" | 5 | 2 | 3 | 58.7 | 55 "Unemployment rate" | 3 | 1 | 2 | 126 |
| 8 "Population" | 6 | 3 | 3 | 90 | 56 "Municipal capacity" | 4 | 4 | 0 | 0 |
| 9 "Natural population increase" | 1 | 0 | 1 | 0 | 57 "Multicurrency" | 1 | 0 | 1 | 0 |
| 10 "Urbanization" | 2 | 1 | 1 | 29 | 58 "Private companies" | 1 | 0 | 1 | 0 |
| 11 "Changes in waste consumption patterns" | 3 | 1 | 2 | 6.7 | 59 "Research" | 3 | 1 | 2 | 67 |
| 12 "Socio-economic class" | 7 | 1 | 6 | 91.5 | 60 "Billed properties" | 2 | 0 | 2 | 0 |
| 13 "Waste collection efficiency" | 5 | 4 | 1 | 316 | 61 "Recyclers" | 1 | 0 | 1 | 0 |
| 14 "Waste collection frequency" | 5 | 4 | 1 | 135.5 | 62 "Loans" | 1 | 0 | 1 | 0 |
| 15 "Harare waste collection policy" | 1 | 0 | 1 | 0 | 63 "Waste collection vehicles" | 3 | 2 | 1 | 117 |
| 16 "Suspension of waste collection services" | 1 | 0 | 1 | 0 | 64 "Workforce capacity" | 6 | 2 | 4 | 253 |
| 17 "Vehicles breakdowns" | 2 | 1 | 1 | 91 | 65 "Fuel" | 3 | 2 | 1 | 28 |

**Table A1.** *Cont.*

| | Degree Centrality | | | Betweenness Centrality | | Degree Centrality | | | Betweenness Centrality |
|---|---|---|---|---|---|---|---|---|---|
| 18 "Vehicles maintenance" | 3 | 2 | 1 | 102 | 66 "Fuel cost" | 1 | 0 | 1 | 0 |
| 19 "Uncollected waste" | 4 | 2 | 2 | 337.5 | 67 "Distance from disposal site" | 1 | 0 | 1 | 0 |
| 20 "Crude waste dumping" | 3 | 2 | 1 | 72.5 | 68 "Community participation" | 1 | 1 | 0 | 0 |
| 21 "Open waste burning" | 2 | 0 | 2 | 0 | 69 "Colonization" | 3 | 0 | 3 | 0 |
| 22 "Waste burying" | 1 | 0 | 1 | 0 | 70 "Travel regulations" | 2 | 1 | 1 | 8 |
| 23 "Waste law enforcement" | 2 | 1 | 1 | 10 | 71 "Illegal settlements" | 2 | 1 | 1 | 56.5 |
| 24 "Receptacle availability" | 4 | 3 | 1 | 42.5 | 72 "Black access to goods and services" | 3 | 2 | 1 | 4.8 |
| 25 "Waste management knowledge" | 4 | 1 | 3 | 28 | 73 "Processed products" | 2 | 1 | 1 | 1.8 |
| 26 "Home tenureship" | 1 | 0 | 1 | 0 | 74 "Racial segregation" | 2 | 1 | 1 | 43.5 |
| 27 "Mosquito and rodent manifestation" | 2 | 1 | 1 | 0 | 75 "Capital city" | 1 | 0 | 1 | 0 |
| 28 "Sewer blockages" | 3 | 1 | 2 | 0 | 76 "Corruption" | 1 | 0 | 1 | 0 |
| 29 "Perceptions" | 5 | 2 | 3 | 32 | 77 "Waste collection fee" | 2 | 1 | 1 | 67 |
| 30 "Street vendors" | 6 | 3 | 3 | 181 | 78 "Communication" | 1 | 0 | 1 | 0 |
| 31 "Environmental pollution" | 2 | 2 | 0 | 0 | 79 "Worker strikes" | 4 | 3 | 1 | 92 |
| 32 "Health risk" | 3 | 3 | 0 | 0 | 80 "PPE" | 1 | 0 | 1 | 0 |
| 33 "Sanitary landfilling operations" | 3 | 2 | 1 | 17.5 | 81 "Salaries" | 2 | 1 | 1 | 51 |
| 34 "Financial capacity" | 13 | 5 | 8 | 268 | 82 "Fights" | 1 | 1 | 0 | 0 |
| 35 "Technical capacity" | 5 | 3 | 2 | 124 | 83 "Sanitary formal solid waste disposal" | 3 | 2 | 1 | 68 |
| 36 "No engineered landfills" | 4 | 2 | 2 | 173.5 | 84 "Waste transfer to poorer suburbs" | 1 | 1 | 0 | 0 |
| 37 "Landfill equipment" | 2 | 0 | 2 | 0 | 85 "Strategy implementation" | 2 | 2 | 0 | 0 |

**Table A1.** *Cont.*

| | Degree Centrality | | | Betweenness Centrality | | Degree Centrality | | | Betweenness Centrality |
|---|---|---|---|---|---|---|---|---|---|
| 38 "Donor funding" | 2 | 1 | 1 | 47 | 86 "Changes in waste composition patterns" | 4 | 4 | 0 | 0 |
| 39 "Economy" | 9 | 6 | 3 | 269 | 87 "HIV/AIDS epidemic" | 1 | 0 | 1 | 0 |
| 40 "Political instability" | 3 | 1 | 2 | 44 | 88 "Cost of service provision" | 4 | 4 | 0 | 0 |
| 41 "Municipal expansion" | 2 | 1 | 1 | 0 | 89 "Accidents" | 1 | 1 | 0 | 0 |
| 42 "Waste recovery" | 5 | 4 | 1 | 60.5 | 90 "Illegal waste dumping" | 14 | 10 | 4 | 300 |
| 43 "Waste separation at source" | 1 | 0 | 1 | 0 | 91 "Population density" | 1 | 1 | 0 | 0 |
| 44 "Waste policy" | 2 | 1 | 1 | 17 | 92 "Dumpsite fires" | 2 | 2 | 0 | 0 |
| 45 "Access control" | 4 | 1 | 3 | 84 | 93 "National informal economy" | 2 | 1 | 1 | 73 |
| 46 "Waste generation rate" | 2 | 1 | 1 | 0 | 94 "National formal economy" | 1 | 0 | 1 | 0 |
| 47 "Disposable income" | 4 | 1 | 3 | 18.5 | 95 "Weather" | 1 | 0 | 1 | 0 |
| 48 "Weighbridge at dumpsite" | 1 | 0 | 1 | 0 | | | | | |

**Table A2.** Network terms and descriptions [12,13,26,27].

| Term | Description | Example in the Study |
|---|---|---|
| Node | An actor in the network. | HSW issues in Harare e.g., population and waste volume. |
| Edge | The link/connection between nodes. | Examples of linked nodes in the network are: Waste recovery—Recyclers Waste recovery—Private companies. |
| Degree centrality | The number of edges directly linked to a node. | Population is directly linked to 7 other nodes. |
| In-degree | The number of nodes directly affecting a given node/the number of nodes directed towards a node. | Population has 3 nodes directly affecting it (edges directed inwards). |
| Out-degree | The number of nodes directly affected by a given node/the number of nodes directed away from a node. | Population has 4 nodes directly being affected by it (edges directed outwards). |
| Betweenness centrality | It is equal to the number of shortest paths from all vertices to all others that pass through the nodes. | It measures the number of times a node acts as a bridge along the shortest path between two other nodes linking nodes that would otherwise not be linked. |

**Table A3.** Harare's waste management issues identified from a literature review conducted by [20].

| Issue | Description | Issue | Description |
|---|---|---|---|
| 1 "Poverty" | The lack of financial resources of the of people to afford a basic standard of living. | 49 "Waste data" | The availability of data that describes waste management in Harare e.g. waste generation rate, waste collection rate etc. |
| 2 "Urban agriculture" | Farming practices in urban areas. | 50 "Planning and monitoring" | The availability of a planning and monitoring services within the waste management department. |
| 3 "Operation Murambatsvina" | Also known as the 'Operation Restore Order', is a government initiative conducted in 2005 where slum areas/illegal structures were forcibly removed across the country. | 51 "Gender" | The sex of the city resident i.e., male or female. |
| 4 "Festive season" | The period leading up to the Christmas and New year holidays. | 52 "Land reform program" | The redistribution of land that occurred in Zimbabwe in the year 2000. |
| 5 "Waste volume" | The quantity of waste produced in Harare. | 53 "Foreign currency" | The available of foreign currency. |
| 6 "Waste collection pressure" | The demand for waste collection services from the local authorities by the city residents. | 54 "Hyperinflation" | The excessive general increase in prices in Harare. |
| 7 "Consumption" | The extent of use of resources by the city residents. | 55 "Unemployment rate" | The increase in the fraction of unemployed people in Harare. |
| 8 "Population" | The amount of people who live in Harare city. | 56 "Municipal capacity" | The ability of the local authorities to handle generated waste. |
| 9 "Natural population increase" | The natural positive change in population in Harare which occurs when live births are higher than deaths recorded in a given period. | 57 "Multicurrency" | The simultaneous use of different currencies in the economy. |
| 10 "Urbanization" | Population shift from rural to urban areas. | 58 "Private companies" | Privately owned companies. |
| 11 "Changes in waste consumption patterns" | - | 59 "Research" | Conducted studies. |
| 12 "Socio-economic class" | Social standing which is measured by place of residence and income. | 60 "Billed properties" | The number of properties that are recognised by the local authorities and hence should pay for municipal services offered. |
| 13 "Waste collection efficiency" | The efficiency with which the local authorities collect waste. | 61 "Recyclers" | The number of people who collect waste for reuse or sale. |
| 14 "Waste collection frequency" | The frequency (i.e. number of waste collection days) with which the local authorities collect waste. | 62 "Loans" | Money borrowed by the local authorities inorder to fund waste management operations. |

Table A3. *Cont.*

| Issue | Description | Issue | Description |
|---|---|---|---|
| 15 "Harare waste collection policy" | The stipulated waste collection frequency by the local authorities. | 63 "Waste collection vehicles" | The number of vehicles available for waste collection. |
| 16 "Suspension of waste collection services" | - | 64 "Workforce capacity" | The number of workers available for waste management as compared to the number of workers needed. |
| 17 "Vehicles breakdowns" | How often waste collection vehicles break down. | 65 "Fuel" | The availability of fuel for waste collection. |
| 18 "Vehicles maintainance" | The efficiency and frequency with which waste collection vehicles are maintained. | 66 "Fuel cost" | The cost of fuel. |
| 19 "Uncollected waste" | The amount of waste that remains uncollected. | 67 "Distance from disposal site" | The distance waste collection vehicles must travel inorder to collect and dispose of waste. |
| 20 "Crude waste dumping" | Waste dumped by unsanitary means at the official Harare dumpsite. | 68 "Community participation" | Involvement of communities in waste management. |
| 21 "Open waste burning" | The waste disposal method where waste is set on fire in open spaces. | 69 "Colonization" | The effect of Zimbabwe's colonization period of 1988–1980. |
| 22 "Waste burying" | The waste disposal method where pits are dug, waste emptied in them and covered in soil. | 70 "Travel regulations" | The restriction of entry of black people in urban areas that was set during the colonial period. |
| 23 "Waste law enforcement" | The efficiency with which waste laws are enforced. | 71 "Illegal settlements" | Non-registered settlements in Harare. |
| 24 "Receptacle availability" | The availability of waste receptacles. | 72 "Black access to goods and services" | During the colonial period. |
| 25 "Waste management knowledge" | Residents' knowledge on how to manage waste. | 73 "Processed products" | Packaged goods. |
| 26 "Home tenureship" | The number of people that own the houses they reside in. | 74 "Racial segregation" | During the colonial period. |
| 27 "Mosquito and rodent manifestation" | The extent to which vermin is prevalent. | 75 "Capital city" | Harare is the capital city of Zimbabwe. |
| 28 "Sewer blockages" | Blockage of sewer pipes which can cause sewer overflows. | 76 "Corruption" | Fraudulent conduct by authorities in the city council. |

**Table A3.** *Cont.*

| Issue | Description | Issue | Description |
|---|---|---|---|
| 29 "Perceptions" | How the city residence view or awareness on waste management issues. | 77 "Waste collection fee" | The waste management service charges for households. |
| 30 "Street vendors" | The number of people illegally selling products in the city streets. | 78 "Communication" | Communication between waste management stakeholders. |
| 31 "Environmental pollution" | The extent of land, water and air pollution. | 79 "Worker strikes" | Strikes by waste management workers. |
| 32 "Health risk" | Exposure of people to the possibility of health endangerment. | 80 "PPE" | Workers' personal protective equipment. |
| 33 "Sanitary landfilling operations" | Waste disposal procedures that follows the environmental, social and public health safety guidelines. | 81 "Salaries" | The amounts of salaries workers receive. |
| 34 "Financial capacity" | The ability of the local government to fund operations. | 82 "Fights" | Conflicts between people. |
| 35 "Technical capacity" | The ability of the local government to offer technical resources needed for waste management operations. | 83 "Sanitary formal solid waste disposal" | Waste disposal that follows the environmental, social and public health safety guidelines. |
| 36 "No engineered landfills" | - | 84 "Waste transfer to poorer suburbs" | - |
| 37 "Landfill equipment" | Equipment required to ensure sanitary landfilling operations. | 85 "Strategy implementation" | Implementation of waste management strategies. |
| 38 "Donor funding" | The amount of external funding received by the local authorities towards waste management in Harare. | 86 "Changes in waste composition patterns" | - |
| 39 "Economy" | National output which determines the goods and services received by the residents. | 87 "HIV/AIDS epidemic" | The effects of the HIV/AIDS epidemic. |
| 40 "Political instability" | Conflicts and competitions between political parties. | 88 "Cost of service provision" | The cost of providing waste management services. |

**Table A3.** *Cont.*

| Issue | Description | Issue | Description |
|---|---|---|---|
| 41 "Municipal expansion" | Positive change in the capacity of the local authorities to manage waste. | 89 "Accidents" | - |
| 42 "Waste recovery" | Reuse of waste either through composting or recycling. | 90 "Illegal waste dumping" | Waste dumped in unsanitary means. |
| 43 "Waste seperation at source" | Amount of waste is separated at source. | 91 "Population density" | The number of people per unit area. |
| 44 "Waste policy" | The policies that govern waste management in Harare. | 92 "Dumpsite fires" | - |
| 45 "Access control" | The control measures set by the local authorities in accessing the official dumpsite. | 93 "National informal economy" | Economic activities and workers that are not registered with the government. |
| 46 "Waste generation rate" | The amount of waste generated per capita per day. | 94 "National formal economy" | Economic activities and workers that are registered with the government. |
| 47 "Disposable income" | The average amount of money available to spend for a family or individual after deduction of taxes. | 95 "Weather" | Rainfall received. |
| 48 "Weighbridge at dumpsite" | The availability of a weighbridge at the official dumpsite. | | |

**Table A4.** Language used to calculate network statistics in RStudio.

| Outcome | Command |
|---|---|
| Generating a Directed Network (kamada kawai layout and circular layout with nodes size at 5) | ○ *library(igraph)*<br>○ *HHSWM_RData<-as.matrix(HHSWM_RData)*<br>○ *HHSWMNetwork<-graph.edgelist(HHSWM_RData, directed = TRUE)*<br>○ *HHSWMNetwork*<br>○ *Plot(HHSWMNetwork)*<br>○ *kamadaLayout<-layout.kamada.kawai(HHSWMNetwork)*<br>○ *plot(HHSWMNetwork, layout = kamadaLayout, vertex.size = 5, vertex.color = "tomato", vertex.frame.color = NA, vertex.label.cex =.7, edge.curved = .1, edge.arrow.size = .3, edge.width = .7)*<br>○ *plot(HHSWMNetwork, vertex.label.cex = .6, vertex.label.color = "black",vertex.color = "tomato", vertex.size = 5, layout = layout_in_circle)* |
| Degree centrality | ○ *degree(HHSWMNetwork, mode = 'all')*<br>○ *degree(HHSWMNetwork, mode = 'in')* |
| Betweenness centrality | ○ *betweenness(HHSWMNetwork, directed = TRUE)* |
| Average path length | ○ *average.path.length(HHSWMNetwork, directed = TRUE, unconnected = TRUE)* |
| Network density | ○ *edge_density(HHSWMNetwork, loops = F)* |
| Network diameter | ○ *diameter(HHSWMNetwork, directed = F, weights = NA)* |
| Network edges | ○ *ecount(HHSWMNetwork)* |
| Network size | ○ *vcount(HHSWMNetwork)* |
| Generating a directed Network (kamada kawai layout and circular layout with node size according to degree centrality and betweenness) | ○ *V(HHSWMNetwork)$degree<-degree(HHSWMNetwork)*<br>○ *plot(HHSWMNetwork, vertex.label.cex = 0.8, vertex.label.color = "black", vertex.size= (V(HHSWMNetwork)$degree/1), layout = layout_in_circle)*<br>○ *V(HHSWMNetwork)$betweenness<-betweenness(HHSWMNetwork, directed = TRUE)*<br>○ *plot(HHSWMNetwork, vertex.label.cex = .8, vertex.label.color = "black",vertex.size= (V(HHSWMNetwork)$ betweenness/25), layout = layout_in_circle)*<br>○ *plot(HHSWMNetwork, vertex.label.cex = .8, vertex.label.color = "black", vertex.size= (V(HHSWMNetwork)$degree/1), layout = kamadaLayout)*<br>○ *plot(HHSWMNetwork, vertex.label.cex = .8, vertex.label.color = "black", vertex.size= (V(HHSWMNetwork)$ betweenness/25), layout= kamadaLayout)* |
| Generating 300 ppi Rplots (e.g degree centrality plot) | ○ *setwd("C:/Users/KWENDA PHYLLIS/Desktop/")*<br>○ *jpeg(file = "RPlot_Degree.jpeg", width = 2300, height = 2000, res = 300)*<br>○ *plot(HHSWMNetwork, vertex.label.cex = .8, edge.arrow.size = .2, edge.color = "green", vertex.label.color = "black",vertex.color = "tomato", vertex.size= (V(HHSWMNetwork)$degree/1))*<br>○ *dev.off()* |

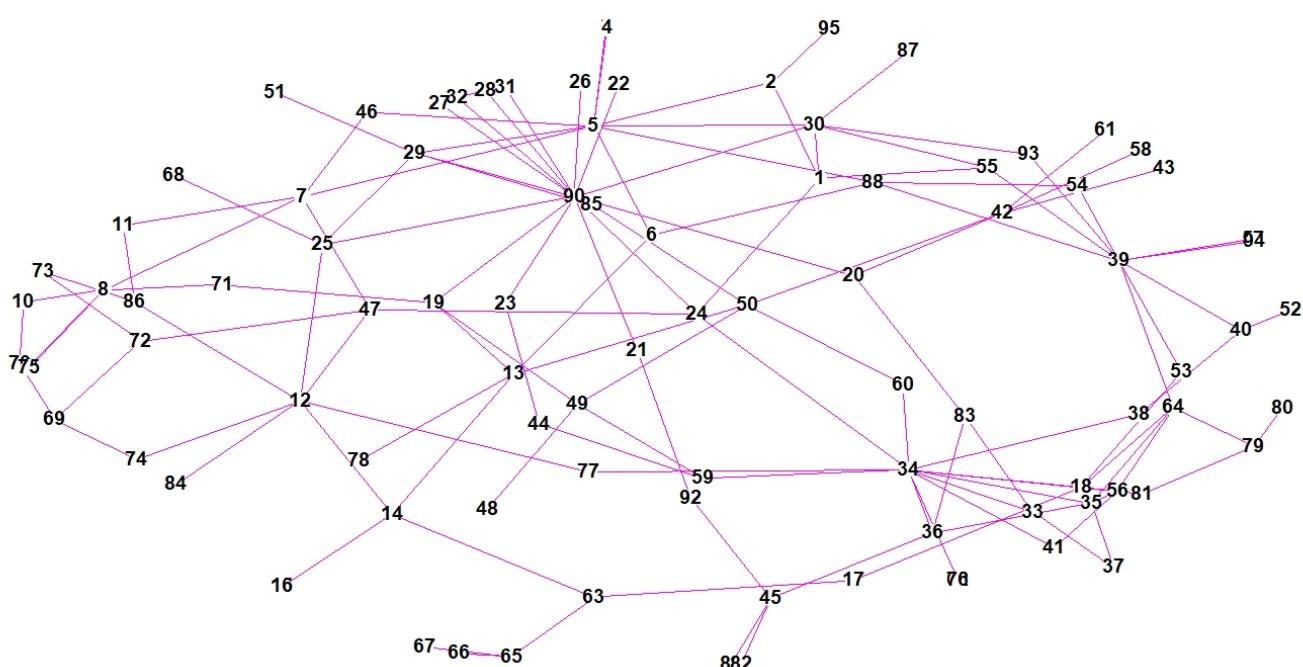

**Figure A1.** Harare's HSW management issues network in Fruchterman-Reingold layout as developed in PAJEK.

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
