# Peer review of "Identifying the Leverage Points in the Household Solid Waste Management System for Harare, Zimbabwe, Using Network Analysis Techniques"

_sustainability, doi:10.3390/su141912405_

Round 1

Reviewer 1 Report

The authors propose: Identifying the leverage points in the household solid waste management system for Harare, Zimbabwe: Using network analysis techniques. I have some concerns and my suggestions are listed below:

1- The contribution is not adequately explained in the abstract. There is no driving force behind the essay. The information was not presented in a way that was understandable and straightforward. The main idea of the work should be emphasized in the abstract section.

2- To highlight research gaps and innovations, the writers should include a Literature Review in the form of tables and concentrate on the study's primary problem in the introductory part.

3- It is important to improve experimental results, validation, and comparison to alternative strategies. More debates and analyses are required.

4- It is crucial to describe the network analysis techniques’ computational complexity.

5- For experiments, nonparametric tests should be used.

Author Response

Thank you for reviewing my manuscript (Manuscript ID: Sustainability-1766809). We are grateful for the constructive comments and suggestions you have provided.

Please find our responses to your review comments.

Kind regards,

Authors.

Reviewer 2 Report

This paper seems to be interesting but there are several flaws that must be addressed to significantly improve the paper in its present form as given in the comments below.

Author Response

(The authors gave the same response as above.)

Reviewer 3 Report

Citations should be presented according to journal guidelines.

Materials and methods are not described in sufficient detail.

The conclusions are too general and do not represent a contribution to the problem of HSW according to the work presented in this investigation.

The authors do not discuss the results from the perspective of previous studies and the result of the applied method "Network analysis using PAJEK, VOSviewer and R soft- 17storage environment". The discussion has presented only the problem of solid waste management. The discussion, as presented in this document, is generic one.

The most important quantitative and qualitative values ​​​​obtained from this research could be included in the conclusions. is one of the authors

It would be useful if you could specify the novelty in the conclusions.

It would be helpful if you could specify the efficiency of the method in the conclusions.

More information, analysis and discussion of the methods and software

used for the analysis of urban solid waste management should be included.

The results obtained through this method can be compared to other methods currently used with similar characteristics.

Author Response

(The authors gave the same response as above.)

Round 2

Reviewer 2 Report

Accept.

Reviewer 3 Report

All of the comments proposed were considered by the authors in the preparation of the final paper.